

# Formation of secondary organic aerosol coating on black carbon particles near vehicular emissions

Alex K. Y. Lee[1], Chia-Li Chen[2], Jun Liu[2], Derek J. Price[2], Raghu Betha[2], Lynn M. Russell[2], Xiaolu Zhang[3, #] and Christopher D. Cappa[3]

[1]Department of Civil and Environmental Engineering, National University of Singapore, Singapore
[2]Scripps Institution of Oceanography, University of California, San Diego, USA
[3]Department of Civil and Environmental Engineering, University of California, Davis, USA
[#]Now at: Crocker Nuclear Laboratory, University of California, Davis, USA

*Correspondence to*: Alex K. Y. Lee (ceelkya@nus.edu.sg)

**Abstract.** Black carbon (BC) emitted from incomplete combustion can result in significant impacts on air quality and climate. Understanding the mixing state of ambient BC and the chemical characteristics of its associated coatings are particularly important to evaluate BC fate and environmental impacts. In this study, we investigate the formation of organic coatings on BC particles in an urban environment (Fontana, California) under hot and dry conditions using a Soot-Particle Aerosol Mass

Spectrometer (SP-AMS). The SP-AMS was operated in a configuration that can detect refractory BC (rBC) particles and their coatings exclusively. Using the $-\log(NO_x/NO_y)$ ratio as a proxy for photochemical age of air masses, substantial formation of secondary organic aerosol (SOA) coatings on rBC particles was observed due to active photochemistry in the afternoon, whereas primary organic aerosol (POA) components were strongly associated with rBC from fresh vehicular missions in the morning rush hours. There is also evidence that cooking related organic aerosols were externally mixed from rBC. Positive

matrix factorization and elemental analysis illustrate that most of the observed SOA coatings were freshly formed, providing an opportunity to examine SOA coating formation on rBC near vehicular emissions. Approximately 7-20 wt% of secondary organic and inorganic species were estimated to be internally mixed with rBC on average, implying that rBC is unlikely the major condensation sinks of SOA in this study. Diurnal cycles of oxygenated organic aerosol (OOA) observed by a co-located standard high-resolution time-of-flight aerosol mass spectrometer (HR-ToF-MS) correlated well with that of SOA coatings on

rBC, but their mass spectral characteristics were different from each other. Our results suggest that at least a portion of SOA materials condensed on rBC surface were chemically different from OOA particles that were externally mixed with rBC, although they are both generated from local photochemistry.

## 1.  Introduction

Black carbon (BC) emitted from incomplete combustion of fossil fuel and biomass has profound impacts on air quality and climate. BC is the dominant absorber of visible solar radiation in the atmosphere, introducing significant contributions to positive radiative forcing on both regional and global scales (Ramanathan and Carmichael, 2008;Bond et al., 2013). Organic coatings can be formed on BC through condensation and/or coagulation of co-emitted primary organic aerosol (POA) and secondary organic aerosol (SOA) produced via photochemical processing. The hydrophilic nature of SOA coating has been

shown to modify hygroscopicity of ambient BC for cloud droplet activation (Kuwata et al., 2009;McMeeking et al., 2011;Laborde et al., 2013;Liu et al., 2013). Increasing coating thickness may enhance light absorption of BC due to a "lensing effect" depending on the degree of particle aging (Jacobson, 2001;Cappa et al., 2012;Peng et al., 2016;Liu et al., 2017), and





alter BC morphology from highly fractal to compact structures and thus their aerodynamic properties (Moffet and Prather, 2009;Schnitzler et al., 2014;Guo et al., 2016;Peng et al., 2016). Understanding the mixing state of ambient BC and the chemical characteristics of its associated coatings is therefore particularly important to evaluate their fate and environmental impacts.

In typical urban environments, traffic emission is one of the major sources of BC particles. A complex mixture of gas-phase organic compounds with a wide range of volatility and molecular structure are co-emitted with BC from vehicles, contributing prominently to the urban SOA burden (Gentner et al., 2017 and references therein). Although it is not straightforward to identify the role of individual SOA precursors, previous studies have shown that organic coating thickness of BC particles and their degree of oxygenation increased with photochemical age or oxidant levels in the atmosphere (Cappa et al., 2012;Liu et

al., 2015;Wang et al., 2017). Of particular concern is the timescale that is required for sufficient SOA condensation to modify the physical, chemical and optical properties of BC near emission sources. Peng et al. (2016) recently performed on-site chamber experiments to examine coating formation on size-selected BC seeds using SOA precursors from particle-free ambient air, demonstrating that only a few hours of photochemical aging can lead to complete particle morphology modification and light absorption enhancement of BC in polluted urban regions. Moffet and Prather (2009) also provided field

evidence that fresh BC can quickly evolve in terms of particle morphology in a photochemically active urban environment by developing coatings of secondary species over a timescale of several hours, highlighting the importance of local SOA chemistry on BC aging mechanisms.

While most previous studies focused on determining potential effects of SOA coatings to the BC properties, there is still a lack

of laboratory and field investigations to examine effectiveness and selectivity of BC seed particles for condensation of SOA materials especially in the presence of other existing seed particles. Recent field observations reported that SOA condensed on BC only accounted for 35% and 41% of total SOA mass near traffic emission sources and in a polluted offshore environment, respectively (Massoli et al., 2012;2015). Metcalf et al. (2013) conducted a series of smog chamber experiments to investigate photooxidation of naphthalene and α-pinene in the presence of both BC and ammonium sulfate seed particles with comparable

surface area of each particle type. Although their observations indicated that the use of BC as a seed is not expected to alter the overall basic chemistry of SOA formation, whether the SOA condensed on BC particles is chemically different from those condensed on ammonium sulfate particles or formed through homogeneous nucleation remains unclear.

Real-time and mass-based chemical compositions of organic coatings on ambient BC particles were seldom reported until the

recent development of an Aerodyne Soot-Particle Aerosol Mass Spectrometer (SP-AMS) (Cappa et al., 2012;Massoli et al., 2012;2015;Onasch et al., 2012;Liu et al., 2015;Lee et al., 2016;Willis et al., 2016). In this study, we investigate formation of organic coatings on BC particles by deploying a SP-AMS in Fontana, California, which is located in the broader South Coast Air Basin and includes the greater Los Angeles area. The sampling site was located in an urban environment with strong influences of vehicular emissions. The SP-AMS was operated in a configuration that can detect refractory BC (rBC) and their

coating materials exclusively. The term of rBC is operationally defined (Onasch et al., 2012) and will be used throughout the rest of this paper. The repeated diurnal patterns of inorganic species, POA and SOA that are internally mixed with rBC reported here provides a unique opportunity to investigate the chemical characteristics and formation of POA and SOA coatings on rBC particles near traffic emissions. A co-located Aerodyne High resolution Time-of-Flight Aerosol Mass Spectrometer (HR-ToF-AMS) was operated simultaneously to quantify the total amounts of non-refractory organic and inorganic speices in $PM_1$

(Chen et al., 2017). The results provide insights into the effectiveness of rBC particles as a condensation sink of fresh SOA near traffic emissions and the chemical characteristics of SOA coatings compared to SOA that were externally mixed with rBC.





## 2. Experiment

### 2.1 Sampling location and instrumentations

The sampling site in Fontana, managed by South Coast Air Quality Management District (SCAQM), was located behind the fire station at 14360 Arrow Highway (34.100 N, 117.490 W). Surrounded by the I-15 freeway to the west (4.3 km), I-10 freeway to the east (3.9 km), and an auto speedway to the south, the sampling site was strongly influenced by vehicular emissions, as well as the broader urban plume. Aerosol particle instruments housed in a sampling van with a custom isokinetic inlet were deployed. Air pulled through the inlet was dried using diffusion driers and subsequently distributed to different real-time particle instruments. This study focuses on the results from a soot-particle aerosol mass spectrometer (SP-AMS, Aerodyne Research) that was configured to detect rBC-containing particles and their coating materials exclusively (See Sections 2.2 and 2.3 for the descriptions of SP-AMS and the calibration approach, respectively). The details of other particle instruments, including a high-resolution time-of-flight aerosol mass spectrometer (HR-ToF-AMS, Aerodyne Research), a single-particle soot photometer (SP2, Droplet Measurement Technologies), and a scanning electrical mobility spectrometer (SEMS, Brechtel) have been reported in Chen et al. (2017) and Betha et al. (2017). Two nitrogen oxide analyzers (Model 42i and 42i $NO_y$, Thermo Fisher Scientific) were used to measure mixing ratios of $NO_x$ and $NO_y$ for determining photochemical age (PCA) of air masses. The heated molybdenum converter in the 42i nitrogen oxide analyzer was replaced by an UV-LED photolytic $NO_2$ converter (Air Quality Design). Hourly average ozone data was obtained from the co-located SCAQM air monitoring station.

Measurements were performed from 5 to 28 July 2015 with ambient temperature varying from 14.9 to 35.9 °C. Companion studies have shown that aerosol compositions were strongly influenced by fireworks from 4 to 8 July 2015 (Chen et al., 2017; Betha et al., 2017). There were few clouds and little precipitation with the exception of a short storm with high rainfall and winds on 18–19 July (two-day precipitation = 3.6 cm and maximum wind speed = 10 ms$^{-1}$). Sampling days with the maximum daily temperature above 30 °C (11–17 and 20–28 July) and lower than 27 °C (9–10 July) were classified as "hot" and "cooler" days, respectively (Chen et al., 2017). The identified hot days were dry with the average hourly relative humidity (RH) varied between 30 and 50% during the daytime. The Hybrid Single-Particle Lagrangian Integrated Trajectory (HYSPLIT) model with input from NOAA Air Resource Laboratory Archived Eta Data Assimilation System was used (http://www.arl.noaa.gov/HYSPLIT_info.php) to evaluate whether particular upwind source regions affected the aerosol measured at the sampling site. Air mass back trajectories were consistently westerly from the coast within the entire sampling period except for the storm days (Chen et al., 2017). Classification of sampling periods based on meteorological conditions and pollutant characteristics is shown in Figure 1.

### 2.2 Soot particle aerosol mass spectrometer (SP-AMS)

The working principle of SP-AMS has been reported in detail previously (Onasch et al., 2012). In brief, rBC-containing particles are vaporized at ~4000 K by a 1064 nm continuous wave intracavity infrared laser similar to that of the SP2 instrument (Onasch et al., 2012). The resulting vapour is ionized via 70 eV electron impact and then detected by a high-resolution time-of-flight mass spectrometer operated in V-mode, which provides a mass resolving power of ~2000 at m/z 28 (DeCarlo et al., 2006; Canagaratna et al., 2007). Note that BC particles detected by SP-AMS are operationally defined as refractory black carbon (rBC). A resistively heated tungsten vaporizer was removed from our instrument so that only rBC and its coating materials were detected (Massoli et al., 2012; 2015; Lee et al., 2016; Willis et al., 2016). An efficient particle time-of-flight system (ePToF, multi-slit chopper with 50% aerosol throughput) was used for measuring aerosol size distributions. The SP-AMS was operated alternating between ensemble mode (i.e., 1-min average of bulk mass spectrum and PToF size distribution) and event trigger mode (i.e., single particle mass spectrum with PToF size).



The SP-AMS was operated from 6 to 28 July 2015 and only the ensemble measurements are reported in this paper. The ensemble data were processed using the AMS data analysis software (Squirrel, version 1.56D for unit mass resolution data and Pika, version 1.15D for high resolution data from http://cires1.colorado.edu/jimenez-group/ToFAMSResources/ToFSoftware/) with the corrected air fragment column of the standard fragmentation table (Allan et al., 2004;DeCarlo et al., 2006). Positive matrix factorization (PMF) was performed to investigate the potential sources and characteristics of rBC and organic aerosol components. The bilinear model was solved using the PMF2 algorithm in robust mode (Paatero and Tapper, 1994) and a final solution was selected using the PMF Evaluation Tool (PET) version 2.06 according to the method described previously (Ulbrich et al., 2009;Zhang et al., 2011). A four-factor solution, including two POA factors from traffic emissions and two SOA factors due to local photochemistry, was selected by examining the solutions for up to eight PMF factors (see supplementary information). Elemental analysis (i.e., oxygen- and hydrogen-to-carbon ratios, O/C and H/C) was performed based on the improved ambient method (Canagaratna et al., 2015a).

### 2.3 SP-AMS calibration

A water suspension of Regal Black (Regal 400R Pigment Black, Cabot Corp., a calibration standard recommended by Onasch et al., 2012), was atomized using a constant output atomizer (TSI Inc., Model 3076) for generating standard rBC particles. Dried 300 nm Regal Black particles were used to determine mass-based ionization efficiency of rBC ($mIE_{rBC}$). Signals for Regal Black particles were quantified by the sum of carbon ion clusters ($C_x^+$, i.e. $C_1^+$-$C_9^+$) using high-resolution mass spectral data. The average $C_1^+$ to $C_3^+$ ratio of 0.478 obtained from Regal Black calibration was used to correct the interference in $C_1^+$ from non-refractory organics in ambient aerosol. The product of material density and the Jayne shape factor (also defined as effective density, $\rho_{eff}$) of the dried 300 nm Regal Black particles was $0.86\pm0.02$ g/cm$^3$, which can be derived from the ratio of the vacuum aerodynamic diameter ($d_{va}$) measured by the SP-AMS to the mobility diameter ($d_m$) selected by a differential mobility analyzer (DMA) as follow:

$$\rho_{eff} = \rho_m\, S = (d_{va} / d_m)\, \rho_0 \qquad \text{(Eq.1)}$$

where $\rho_m$ and $S$ are the material density and Jayne shape factor, respectively, and $\rho_0$ is the unit density (DeCarlo et al., 2004). The average effective density of 0.86 and $d_m$ were used to further calculate the mass of individual dried 300 nm Regal Black particles (DeCarlo et al., 2004) which was approximately 12.2 fg. The average $mIE_{rBC}$ value was $255\pm50$ ions/pg of Regal Black particle based on three independent calibrations performed throughout the study.

Direct calibration of the ionization efficiency for nitrate ($IE_{NO3}$) is not possible without the tungsten vaporizer. Before removal of the tungsten vaporizer from the SP-AMS, dried 300 nm pure ammonium nitrate ($NH_4NO_3$) and Regal Black particles were generated for determining mass-based ionization efficiency of nitrate ($mIE_{NO3}$) and $mIE_{rBC}$, respectively. Note that the $mIE_{NO3}$ was determined without operating the laser vaporizer. The relative ionization of rBC ($RIE_{rBC} = mIE_{rBC}/mIE_{NO3}$) was 0.26. Assuming that $RIE_{rBC}$ remains unchanged after removing the tungsten vaporizer, $mIE_{NO3}$ and $IE_{NO3}$ were calculated based on measured values of $mIE_{rBC}$. The calculated $IE_{NO3}$ was then used with recommended RIE values (Jimenez et al. (2003), i.e., nitrate = 1.1, sulfate = 1.2, chloride = 1.3, organics = 1.4 and ammonium = 4) to quantify non-refractory aerosol species associated with rBC (referred to as NR-PM$_{rBC}$). Note that our previous studies have shown that this calibration approach likely yields the calculated $IE_{NO3}$ values as a lower limit, leading to over quantification of the NR-PM$_{rBC}$ mass loadings in ambient aerosol (Willis et al., 2014;Lee et al., 2015).



The collection efficiency (CE) for rBC particles that is governed by the degree of overlap between particle and laser beams was determined using beam width probe (BWP) measurements described previously (Willis et al., 2014). Ambient rBC-containing particles had an average beam width ($\sigma$) = 0.45±0.04 mm based on three sets of BWP measurements performed throughout the study. The measured particle beam width suggests a condition of incomplete beam overlap, arising from non-spherical rBC particles, and hence a CE of 0.6 was applied for absolute quantification of rBC and NR-PM$_{rBC}$ (Willis et al., 2014). All BWP measurements were performed around the morning rush hours. Fresh rBC-containing particles from vehicular emissions in the morning had thinner coatings compared to those rBC-containing particles observed in the afternoon that were more photochemically aged (see discussion in Section 3). It has been demonstrated that CE for rBC particles increases (or rBC particle beam width decreases) with coating thickness (Willis et al., 2014). The applied CE may be therefore less relevant for the time with high ambient SOA loading (i.e., high NR-PM$_{rBC}$/rBC mass ratio, or R$_{BC}$). Nevertheless, the CE applied does not impact calculations of R$_{BC}$ in the rBC-containing particles.

## 3. Results and Discussion

Figure 1 shows the time series of meteorological data (temperature, relative humidity (RH), wind direction and wind speed), mixing ratios of ozone and NO$_x$, NO$_x$/NO$_y$ ratio, and chemical compositions of rBC-containing particles (i.e., rBC and NR-PM$_{rBC}$) from 5 to 28 July 2015. The whole sampling period can be divided into four categories as previously reported (Chen et al., 2017; Betha et al., 2017). In brief, fireworks had strong impacts on aerosol compositions from 5 to 8 July, and significant enhancements of inorganic aerosol components were observed in rBC-containing particles (Figure 1f). After the firework period, mass loadings of rBC-containing particles remained low from 9 to 10 July without clear diurnal patterns (indicated as cooler days in Figure 1). A storm cleaned up the atmosphere from 18 to 19 July, leading to very low levels of all aerosol components within the two days. The weather was relatively hot and dry for the rest of sampling days (indicated as hot days in Figure 1). Repeated diurnal patterns of Org/rBC ratio (or R$_{BC}$) with peak values in the afternoon were observed over the hot period (Figure 1e), indicating a unique opportunity to examine POA and SOA formation on rBC near vehicular emissions via daytime chemistry. The observations during the hot period will be the focus of the following discussion.

### 3.1 Chemical characteristics of rBC coating materials (NR-PM$_{rBC}$)

Figure 2a shows the average diurnal cycles of rBC and NR-PM$_{rBC}$ during the hot period. Mass loadings of rBC increased continuously in the morning rush hours and peaked at ~8:00–9:00, and they were strongly correlated with the NO$_x$ mixing ratio (r = 0.82), consistent with local traffic emissions as a major source (Figure 1d). Boundary layer break up led to the decrease of rBC and NO$_x$ concentrations (i.e., dilution as air from residual layer mixed down to the surface). These diurnal patterns indicate that there were minimal influences of other combustion sources such as biomass burning and industrial emissions to the observed rBC concentrations. Nitrate and ammonium concentrations correlated well with each other (r = 0.97). Their mass loadings increased slowly over the night and reached the maximum levels at ~10:00–11:00 in the morning (Figure 2a). While NO$_2$ reacts with ozone to generate N$_2$O$_5$ at night, OH radicals oxidation of NO$_2$ during daytime produce nitric acid that can be neutralized by ammonia, forming particulate NH$_4$NO$_3$. A recent tunnel study has observed such NH$_4$NO$_3$ formation chemistry in aged traffic emissions (Tkacik et al., 2014). The decrease of particulate nitrate and ammonium (i.e., NH$_4$NO$_3$) concentrations could be the combined effects of boundary layer break up and evaporative loss at the increasing temperature during the day. Although sulfate remained in relatively low concentrations in both day and night time, it followed the diurnal patterns of nitrate in the morning (see also Figure 5d), indicating significant contributions from local secondary aerosol formation chemistry. Chloride was also associated with traffic (see also Figure 5e) and was likely due to condensation of HCl vapor that was subsequently neutralized by ammonia (i.e., formation of NH$_4$Cl).



Organic aerosol (OA) was the dominant component of NR-PM$_{rBC}$ during the hot period as illustrated in Figure 2a. Mass loadings of organic coating increased with rBC concentrations in the morning rush hours. PMF results show that two primary emission factors, referred to as hydrocarbon-like OA-rich (HOA-rich) and rBC-rich factors hereafter, were the major

contributors to the total OA mass during the morning rush hours (Figure 3e and f). The mass concentration of the rBC-rich factor was slightly lower than the HOA-rich factor. Mass spectra of HOA- and rBC-rich factors indicate that rBC accounted for 14.2 and 44.4 wt% of the two primary aerosol factors, respectively. The fragmentation pattern of rBC was similar to those previously reported in urban environments near traffic emissions and engine exhausts (Massoli et al., 2012;Lee et al., 2015;Enroth et al., 2016;Willis et al., 2016;Saarikoski et al., 2017). With the support of single particle measurement, Willis et

al. (2016) also separated traffic-related OA into HOA- and rBC-rich factors in the roadside environment using a SP-AMS with a tungsten vaporizer being removed and estimated that approximately 90% of rBC mass emitted from vehicle resided in rBC-rich particles. By following the calculation procedure described in Willis et al. (2016), rBC-rich factor contributed about 82 wt% of the freshly emitted rBC from traffic.  Note that rBC-rich particles were composed of more oxygenated organic fragments compared to the HOA-rich factor (Figure 3a and b), likely due to the presence of refractory ion fragments (i.e., $CO^+$

and $CO_2^+$) that originated from oxygenated functionalities on the soot surface and in the soot nanostructure (Corbin et al., 2014;Malmborg et al., 2017).

Using $-\log$ (NO$_x$/NO$_y$) as a proxy for PCA of air masses and Org/rBC ratio (or R$_{BC}$) as an indicator for SOA formation, production of fresh SOA coating materials on rBC particles was observed due to active photochemistry in the afternoon (Figure

2b). The secondary nature of organic coatings observed in the afternoon peak was supported by the diurnal cycles of O/C, H/C and average carbon oxidation state (OS$_c$ ≈ 2O/C – H/C, Kroll et al. (2011)) of total OA (Figure 2c). Higher values of O/C and OS$_c$ were observed in the afternoon compared to in the morning rush hours, consistent with the expectation that the O/C ratio of SOA is greater than POA. The PMF analysis identified two types of oxygenated OA (OOA), referred to as OOA-1 and OOA-2 hereafter, both of which are likely SOA coatings. Mass spectra of both OOA factors were dominated by an oxygen-

containing organic fragment at m/z 43 (i.e., $C_2H_3O^+$, Figure 3c and d), similar to those previously observed by Massoli et al. (2015). OOA-2 represented more oxygenated fraction of SOA coating (O/C = 0.62) with the maximum mass loadings observed at ~15:00-16:00 and its diurnal pattern matched well with Org/rBC ratios (Figures 2b and 3h). OOA-1 represented less oxygenated SOA components (O/C = 0.53) and, in addition to the afternoon peak, OOA-1 increased in the late morning coincident with nitrate and ammonium (i.e., peaks at ~10:00-11:00, Figure 2a and 3g). This suggests a possibility that OOA-1

represented a fresher portion of SOA coating materials generated by photochemistry of anthropogenic gas-phase precursors from vehicular emissions.

## 3.2  Chemical characteristics of OA as a function of coating thickness (R$_{BC}$)

Figure 4a illustrates that R$_{BC}$ increased continuously as a function of $-\log$ (NO$_x$/NO$_y$) within the hot period. Assuming ambient

daytime OH radical concentration was ~4 × 10$^6$ molecules cm$^{-3}$ (Takegawa et al., 2006;Slowik et al., 2011) and the major NO$_x$ loss product was HNO$_3$, the estimated PCA values (i.e., PCA ≈ $-\ln$ ([NO$_x$]/[NO$_y$]) / k$_{rxn}$ [OH]) were about 5-7 h in the afternoon given that the rate constant between OH radicals and NO$_x$ for HNO$_3$ formation (k$_{rxn}$) is equal to 7.9 × 10$^{-12}$ cm$^3$ molecules$^{-1}$ s$^{-1}$ (Brown et al., 1999;Cappa et al., 2012). Such estimation further supports our hypothesis that OOA-1 and OOA-2 were fresh SOA produced in the local atmosphere. Furthermore, OA components became more oxygenated as the PCA and R$_{BC}$ increased

(i.e., solid circles with the coloured scale of OS in Figure 4a). Figure 4c illustrates that ~90 wt% of total NR-PM$_{rBC}$ was organic regardless of the values of R$_{BC}$. POA from traffic emissions (i.e., rBC- and HOA-rich factors) accounted for ~60–75 wt% of



total NR-PM$_{rBC}$ when R$_{BC}$ is smaller than 4 whereas the contributions of SOA components (i.e., OOA-1 and OOA-2 factors) increased with R$_{BC}$, reaching a plateau at about 70–75% of total NR-PM$_{rBC}$ when R$_{BC}$ is larger than 8.

Figure 4b shows that R$_{BC}$ decreased continuously with higher rBC loadings, highlighting the fact that most of the rBC mass observed within the hot period was associated with POA materials. Based on the mass fraction of rBC signals in the mass spectra of each PMF factor, it can be estimated that over 80 wt% of rBC mass was associated with POA (i.e., about 60 and 20 wt% from rBC-rich and HOA-rich factors, respectively) when R$_{BC}$ is smaller than 4 (Figure 4d). An increasing contribution of OOA factors to rBC mass was observed for particles with thicker OA coating. OOA-1 contributed up to ~60 wt% of rBC mass when R$_{BC}$ is larger than 10 while OOA-2 was only a minor contributor to rBC mass for the whole range of R$_{BC}$ values. The small contribution of OOA-2 particles to the rBC burden occurred despite the substantial contribution of OOA-2 to the total NR-PM$_{rBC}$ mass. This is because rBC accounted for only 0.5 wt% of the OOA-2 factor, implying that such OOA materials co-existed with small rBC inclusions. Willis et al. (2014) reported that SP-AMS could accurately measure the mass fraction of rBC, at least down to 0.05 (5 wt%), in laboratory-generated organically coated Regal Black particles. Vaporization efficiency of an individual particle with a tiny rBC core diameter (e.g., < 5 wt%) and its uncertainties to mass quantification remain unclear (e.g., insufficient volatilization may lead to an underestimate of mass in the factor).

### 3.3 Comparisons of NR-PM$_{rBC}$ and NR-PM components

To understand the mixing state of OA and rBC particles, the SP-AMS measurements (NR-PM$_{rBC}$) were compared to the co-located HR-ToF-AMS measurements (NR-PM) as presented in Figures 5 and 6. Mass loadings of secondary species in NR-PM$_{rBC}$ were lower than their corresponding NR-PM components based on the CE and IE$_{NO3}$ values used in this work, suggesting that significant fractions of secondary aerosol species were externally mixed with rBC. Specifically, diurnal cycles of nitrate, ammonium and chloride were strongly correlated (r > 0.96) between the two measurements but only about 8-20 wt% (or NR-PM$_{rBC}$/NR-PM = 0.08-0.2) of their masses were coated on rBC. The fraction of sulfate that was internally mixed with rBC was small, on average (NR-PM$_{rBC}$/NR-PM = 0.07). The relative abundance of HOA and OOA (i.e., comparing OA mass loadings in the morning and afternoon peaks in Figure 5a) suggests that a larger fraction of OOA was externally mixed with rBC compared to HOA from traffic emissions, discussed further below. In general, OOA and sulfate tend to be concentrated in the accumulation mode peaking between 400-600 nm in d$_{va}$ (Zhang et al., 2011), and hence the above argument also explains the larger difference of OA and sulfate mass with d$_{va}$ larger than 300 nm between the two measurements (Figure 5i).

Chen et al. (2017) identified four PMF factors, namely HOA, cooking OA (COA), nitrate-related OOA (NOOA) and vehicle-related OOA (VOOA), to describe the potential sources of OA measured by the HR-ToF-AMS (Figure S6) in this field campaign. The HOA and COA factors are assumed primary and the NOOA and VOOA factors are assumed secondary in origin. First, HOA exhibited a strong peak in the morning rush hours, and its diurnal cycle and mass loading was very similar to the sum of rBC- and HOA-rich factors in terms of both shape and the absolute concentrations (Figures 5f). HOA accounted for 11% of total OA in NR-PM. Figure 6 further demonstrates that the average ratio of NR-PM$_{rBC}$/NR-PM for HOA components (i.e., ([HOA-rich] + [rBC-rich]) / [HOA]) is about 0.95. Note that the mass loadings of rBC contributing to the HOA-rich and rBC-rich factors (estimated from the C$_n^+$ ions) were subtracted in the calculation of NR-PM$_{rBC}$/NR-PM. Nevertheless, the presence of refractory oxygenated organic fragments in the mass spectra of the rBC-rich factor could introduce positive biases to such estimation. The average ratio drops to about 0.85 if the three major oxygenated organic fragments, including CO$^+$, CO$_2^+$ and C$_2$H$_3$O$^+$, are also excluded in the calculation. Using the same measurement approach,




Massoli et al. (2012; 2015) reported that 81 and 87 % of HOA were associated with rBC particles near vehicular emissions and in a polluted offshore environment.

COA was another POA that contributed to 15 wt% of total OA in NR-PM. However, COA was not identified in the PMF
analysis of SP-AMS data. The comparison suggests that COA was unlikely co-emitted with rBC from modern kitchens, and the mixing of rBC and COA through particle coalescence was insignificant near the sampling location. Similar observations have been reported in previous studies in downtown Toronto. Willis et al. (2016) could not identify COA by measuring rBC-containing particles exclusively whereas Lee et al. (2015) could separate a COA factor from other OA components by deploying a SP-AMS equipped with dual vaporizers (i.e., laser and tungsten vaporizers). Comparing the results obtained from
two different operational modes (i.e., switching laser vaporizer on and off), Lee et al. (2015) provided indirect evidence that COA was largely externally mixed with rBC in the urban atmosphere.

SOA was the most dominant OA component. VOOA and NOOA accounted for 64 and 10 wt% of total OA in NR-PM, respectively. Even though both OOA-2 and VOOA were formed through local photochemistry in the afternoon (Figure 5h),
VOOA mass concentrations started dropping substantially at ~18:00, which was about 1-2 hours delay compared to the case of OOA-2. This observable delay corresponded to the time of increasing sulfate levels in NR-PM, implying additional sources and formation pathways of VOOA that might be related to the transport and formation chemistry of sulfate in NR-PM. VOOA might form in a downwind environment with less rBC loadings and advect to the site when the OOA-2 concentration dropped. The diurnal cycle of NOOA strongly correlated with nitrate and ammonium in NR-PM (r > 0.93) whereas OOA-1 exhibited a
bimodal pattern that was similar to sulfate in NR-PM$_{rBC}$ (Figure 5g). The bimodal pattern suggests that the formation mechanisms of OOA-1 in the morning and afternoon may be slightly different but they cannot be separated by PMF due to their chemical similarity. Assuming a certain fraction of OOA-1 was produced through photochemistry by following the diurnal pattern of OOA-2, mass loadings of OOA-1 in the morning and afternoon peaks can be roughly estimated as illustrated by the red dashed lines in Figure 5g. The estimated OOA-1 peak in the morning matches reasonably well with the NH$_4$NO$_3$
and NOOA peaks, suggesting OOA-1 observed in the morning may be related to nitrate formation chemistry. Furthermore, the estimation yields that the production rate of OOA-1 in the afternoon was comparable to that of OOA-2 and the peak mass loadings of VOOA were about 5 times higher than the total of OOA-1 and OOA-2 (Figure 5g and 5h). Overall, approximately 20 wt% of OOA components were estimated to be internally mixed with rBC on average (Figure 6).

A key question remaining is whether the OOA materials identified by the SP-AMS and HR-ToF-AMS are the same in terms of AMS mass spectral characteristics. Figures 3 and S6 show that the mass spectra of OOA factors measured by the two instruments were clearly distinct from each other. Specifically, VOOA and NOOA were dominated by an organic fragment of CO$_2^+$ (i.e. a tracer of organic acids) and CH$_2$O$^+$, respectively whereas C$_2$H$_3$O$^+$ was the major fragment of OOA-1 and OOA-2. However, it is particularly important to point out that different aerosol vaporization schemes utilized in SP-AMS and HR-ToF-
AMS makes the direct comparison of organic mass spectra not straightforward. It has been demonstrated that thermal vaporization (operated at 600ºC) used in the HR-ToF-AMS produces significant decomposition and dehydration of oxidized organic compounds (Canagaratna et al., 2015a) but the laser vaporization used in SP-AMS can provide soft vaporization of organic coatings on rBC particles at lower temperature, resulting in less molecular fragmentation (Canagaratna et al., 2015b). The OOA-1 and OOA-2 spectra notably have more peaks and with higher intensities at larger m/z (> 60 amu) compared to the
NOOA and VOOA spectra. Also, the relative intensity of the peaks at m/z 28, 29 and 30 (CO$^+$, CHO$^+$ and CH$_2$O$^+$, respectively) are substantially reduced in the OOA-1 and OOA-2 spectra compared to NOOA and VOOA. Both of these observations are consistent with reduced fragmentation from vaporization in the SP-AMS being a major reason for the differences.



The elemental ratios (O/C and H/C) extracted from the SP-AMS and HR-ToF-AMS mass spectra of oxidized organic species have been shown to be different. Canagaratna et al. (2015b) reported that the SP-AMS O/C and H/C values differ from their corresponding HR-ToF-AMS values by factors of 0.83 and 1.16, respectively, based on the laboratory analysis of chemical standards, including dicarboxylic acids, multifunctional acids and alcohols. These conversion factors are applied to the O/C and H/C ratios of NOOA and VOOA (Chen et al., 2017) in order to perform a more meaningful comparison to our SP-AMS measurements in the Van Krevelen diagram (Figure 7). The O/C and H/C ratios of OOA-2 are similar to those of VOOA (0.62 vs. 0.66 and 1.63 vs. 1.67, respectively), well within the measurement uncertainties. The O/C of OOA-1 is lower than the adjusted O/C of either NOOA or VOOA, suggesting some chemical difference between OOA-1 and the externally mixed SOA. However, the non-weighted average of the adjusted H/C for NOOA and VOOA ($H/C_{avg}$ = 1.97) is very similar to that of OOA-1 (H/C = 1.93). More field and laboratory data are required to validate and improve the empirical relationships proposed by Canagaratna et al., (2015b) and to understand the extent to which the observed differences are a result of true chemical differences versus explainable by differences in molecular fragmentation due to the different vaporization schemes used in the instruments.

## 4. Conclusions and Atmospheric Implications

The repeated diurnal patterns of inorganic species, POA and SOA reported here provide a unique opportunity to investigate the chemical characteristics and formation of OA coating on rBC particles near traffic emissions. There is no doubt that HOA was internally mixed with rBC significantly as they were largely co-emitted by vehicles. The results of PMF illustrate that rBC- and HOA-rich factors accounted for about 60 and 20 wt% of rBC with thin coating, respectively, and the rBC-rich factor contributed about 82 wt% of the freshly emitted rBC from traffic, similar to previous observation in the roadside environment (Willis et al., 2016). The COA factor is commonly observed in urban areas (Allan et al., 2010;Mohr et al., 2012) but its mixing with ambient rBC is seldom reported. The absence of a COA factor in rBC-containing particles highlights the fact that emissions of rBC from modern kitchens and the mixing of rBC and COA through particle coalescence were negligible in this study. Previous measurements conducted in an urban area also pointed to the same conclusion (Lee et al., 2015;Willis et al., 2016).

Increases in coating thickness were primarily due to substantial formation of fresh SOA through local photochemical processing on the timescale of a few hours. On average, about 7-20 wt% of secondary aerosol species, including both inorganic and OOA species, were condensed on rBC particles, suggesting that rBC was unlikely the major sink for condensation of fresh SOA in this study. During the peak of SOA production, the average mass loadings of rBC and VOOA were about 0.2 and 6.8 $\mu g/m^3$, respectively, which were a few factors to orders of magnitude lower than those generated in some recent aging experiments of soot particles (Metcalf et al., 2013;Li et al., 2017). Peng et al. (2016) recently showed that the timescale for producing sufficient fresh SOA to completely modify rBC properties strongly depends on pollution levels. Our observations may provide insight into the design of soot aging experiments for investigating the formation rate of fresh SOA coatings (e.g., growth rate of coating thickness) as well as their environmental impacts under a more atmospherically relevant condition.

Our measurement approach leads to a conclusion that at least a fraction of OOA condensed on rBC was chemically distinct from that externally mixed with rBC, although uncertainties of organic fragmentation due to the application of the laser vaporizer still need to be fully established to quantify this difference. The reason for this unique observation remains unclear. One possibility is that SOA production via photooxidation of anthropogenic volatile organic compounds (VOCs) started occurring near traffic emission sources in which relatively high concentrations of rBC particles played a more critical role for



SOA condensation compared to other existing background particles. In particular, certain classes of SOA precursors may oxidize more rapidly or require less oxidation steps (e.g., Intermediate VOCs vs. VOCs) to low-volatility organics than the others, forming a different type of SOA near emissions and further downfield. Atmospheric dilution modified the chemical compositions and concentrations of SOA precursors and seed particles so that the formation of coatings on rBC might be less

efficient and chemically different under the diluted conditions. For example, VOOA formation in the late afternoon linked to the transport and formation chemistry of inorganic sulfate, which was probably less relevant to the formation of OOA coating on rBC. Research efforts are required to examine whether certain SOA materials are more favorable to condense on rBC-containing particles compared to other types of seed particles. The roles of hydrophobic coating (e.g. HOA) and soot surface functionality on SOA condensation need to be further explored.

**Acknowledgements**

Lee A. K. Y. would like to acknowledge the support from the NUS Start-up Grant (R-302-000-173-133). Cappa D. C. and Russell L. M. were supported by the California Air Resources Board (Agreement number: 13-330)The authors would also like to thank the Fontana Fire Department and the South Coast Air Quality Management District (SCAQMD) for Fontana site logistics.

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

45



**Figures**

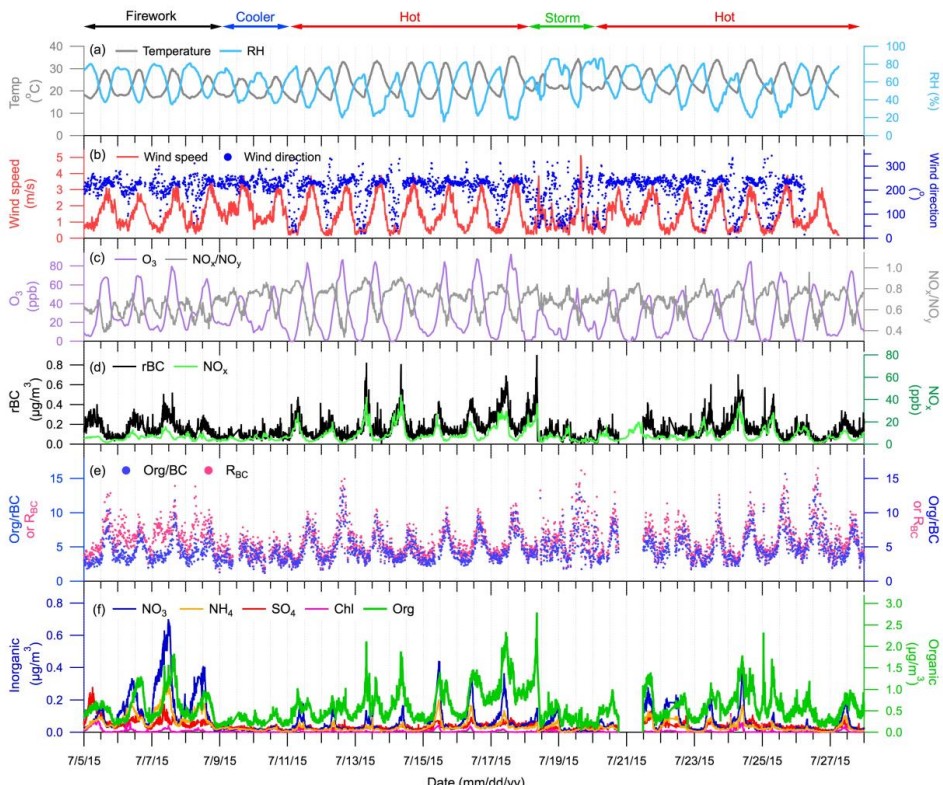

Figure 1: Time series of (a) temperature and RH, (b) wind speed and direction, (c) ozone and $NO_x/NO_y$ ratio, (d) rBC and $NO_x$,

5 (e) Org/rBC ratio and $R_{BC}$, and (f) NR-PM$_{rBC}$ (NO3 = nitrate, NH4 = ammonium, SO4 = sulfate, Chl = chloride and Org = organic)



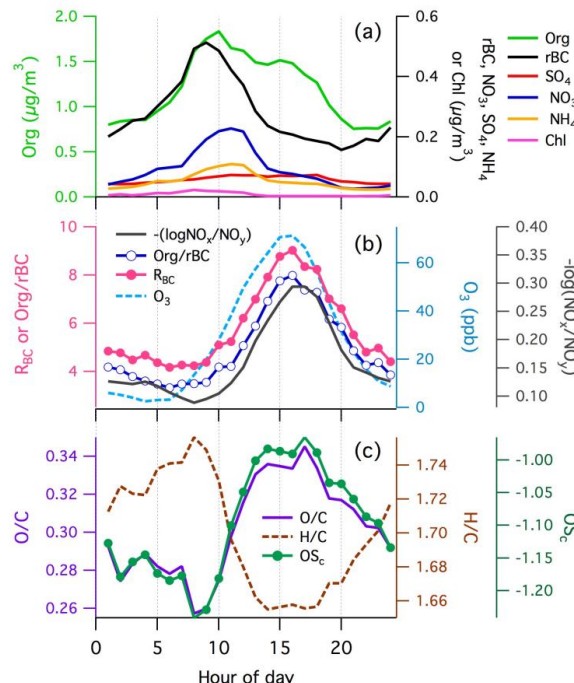

Figure 2: Diurnal cycles of (a) rBC and NR-PM$_{rBC}$, (b) Org/rBC ratio, R$_{BC}$, ozone and –log (NO$_x$/NO$_y$), and (c) O/C, H/C and OS within the hot period



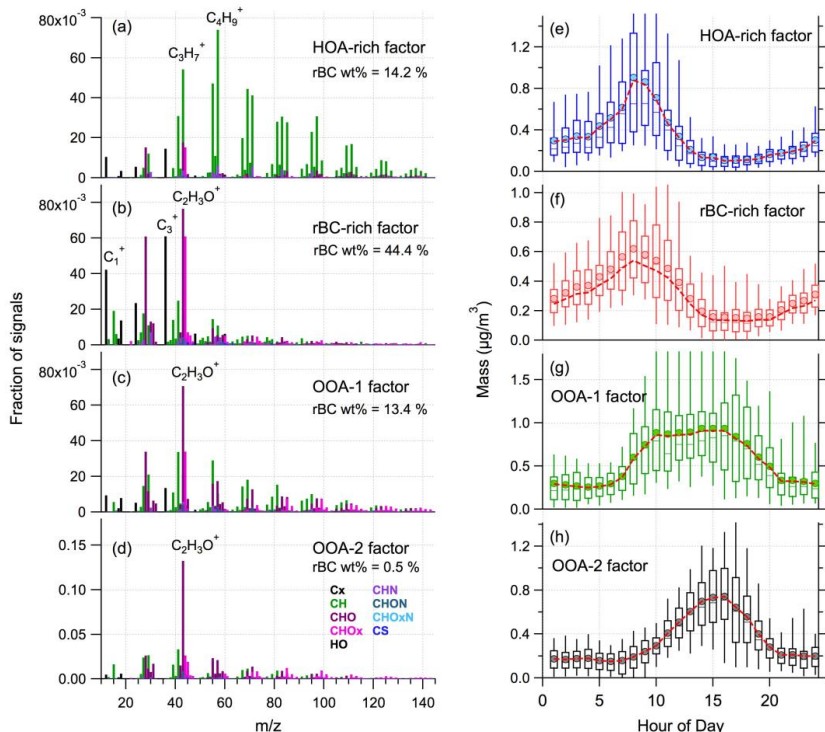

Figure 3: Mass spectra (a-d) and diurnal cycles (e-h) of PMF factors from SP-AMS data within the hot period. (Box plots: 5th,
25th, 50th, 75th and 95th percentile, Filled circles: mean values for organic + $C_x^+$ fragments, Red dashed lines: mean values for
5   organic alone)



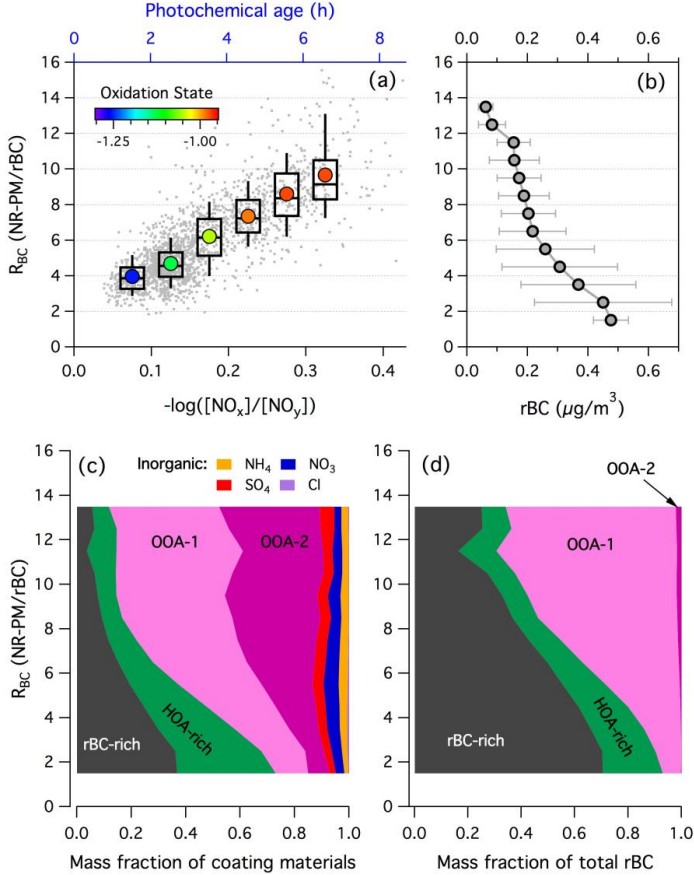

Figure 4: Coating thickness ($R_{BC}$) as a function of (a) photochemical age, (b) rBC mass loadings, (c) chemical compositions of coating, and (d) rBC mass fractions contributed by individual PMF factors within the hot period



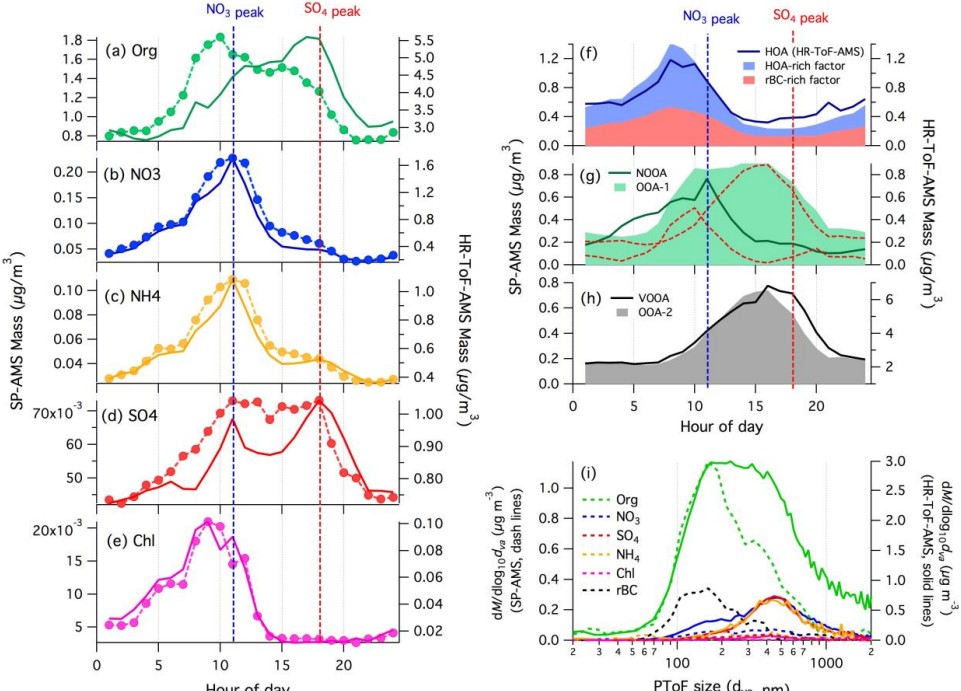

Figure 5: (a-e) Diurnal cycles of NR-PM and NR-PM$_{rBC}$ measured by HR-ToF-AMS (solid lines) and SP-AMS (dashed lines with circles), respectively. (f-h) Diurnal cycles of PMF factors from HR-ToF-AMS (solid lines) and SP-AMS (filled areas) data. (i) PToF size distribution of rBC, NR-PM (dashed lines) and NR-PM$_{rBC}$ (solid lines). Mass loadings of OOA-1 in the morning and afternoon peaks can be roughly estimated as illustrated by the red dashed lines in panel g assuming OOA-1 was produced through photochemistry by following the diurnal pattern of OOA-2 in the afternoon.





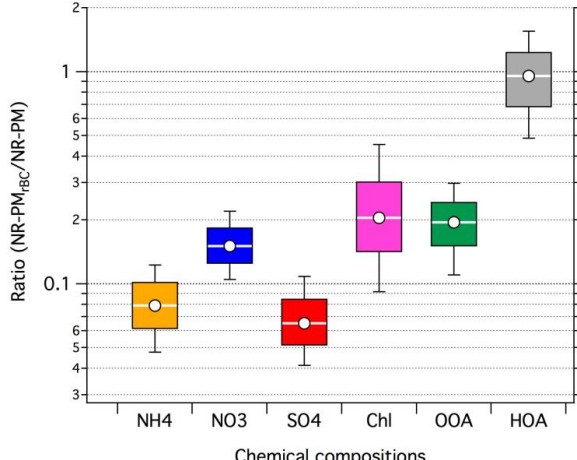

Figure 6: NR-PM$_{rBC}$-to-NR-PM ratios for individual aerosol components and PMF factors (Box plots: 10[th], 25[th], 50[th], 75[th] and
90[th] percentile, White circles: mean values, OOA = ([OOA-1] + [OOA-2]) / ([NOOA] + [VOOA]) and HOA = ([HOA-rich] +
[rBC-rich]) / [HOA]).





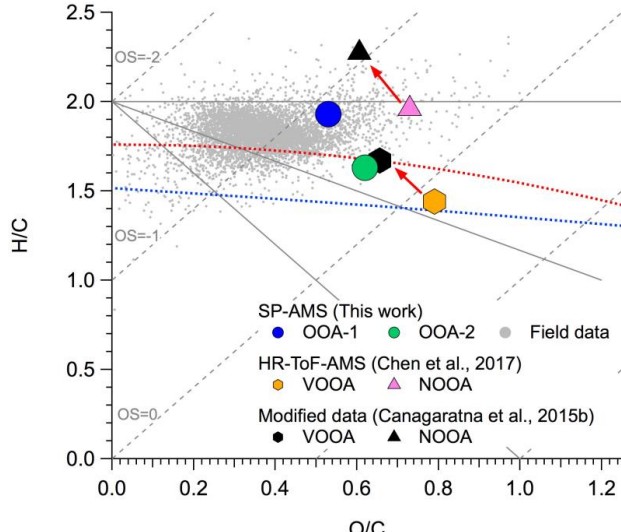

Figure 7: Van Krevelen diagram: Red arrows indicate the changes in the elemental ratios of VOOA and NOOA factors after
5  applying the correction factors for more oxygenated organic species proposed by Canagaratna et al. (2015b).