# Peer review of "Formation of secondary organic aerosol coating on black carbon particles near vehicular emissions"

_Atmospheric Chemistry and Physics, 2017_

## Referee Comment (RC1) · Anonymous Referee #1 · 28 Aug 2017

General comments

This paper compares SP-AMS and AMS data at an urban location in the US to infer details concerning the behaviour of black carbon aerosols and SOA formation in this environment. This is partly facilitated by the dataset exhibiting a strong diurnal profile, meaning that traits can be reported with decent statistical power, in spite of the sampling period being relatively short. While the quantitative accuracy of the SP-AMS may be questionable (many of the RIEs and CEs are simply assumed rather than calibrated here), this does not form the major part of the science. This paper is well written and has the potential to provide some useful insight relevant to ACP, however I feel that the discussion side of the paper needs more work. Specifically, the authors appear to be making two unsafe assumptions often inappropriately made when analysing data

of this nature; firstly, they seem to be assuming that the patterns are governed by sources and in situ processes (neglecting advection), akin to a box model and second, they seem to be treating the PMF factors as discrete chemical entities. While I'm sure the authors are aware of the issues here, this is not currently reflected in the text and a few conclusions are made that I would regard as unsafe. I recommend that this be published subject to major corrections.

Specific comments

There an issue that I can see regarding the conclusions of the comparison of organic factors from the two AMSs in that the authors appear to be treating the PMF-derived SOA factors as discrete chemical types. While this could hypothetically be the case, it is far more likely that when applying the simple PMF data model to a chemically complex system such as this, the two SOA factors for each instrument represent the mathematical endpoints of a distribution of data, within the non-negativity constraint imposed. This being the case, a direct comparison of the two instruments' factors may be problematic; given that it is known their mass spectral responses to different organic materials are fundamentally different, their respective factorisations could end up being very different in character for the same aerosol simply through virtue of how the spectra are distributed in multivariate space and what endpoints are converged on by the solver. As a suggestion, a more objective comparison of the two datasets might be achieved if the primary factors (determined using PMF) are subtracted from the data matrices then metrics such as elemental ratios derived from what is left (assuming it is all secondary), e.g. as new points on figure 7. See https://www.atmos-chem-phys.net/15/6351/2015/.

An entire line of discussion – comparing the behaviour of organics to sulphate – I consider to be deeply flawed in the way it has been approached here. Outside of aqueous-phase processing (which is almost certainly not happening here, given the low humidities), the formation lifetime of sulphate from the gas phase oxidation of sulphur dioxide tends to occur on the timescale of weeks (far beyond the timescale where

NOy/NOx is a useful indicator of photochemistry), which means that a diurnal pattern observed here is extremely unlikely to be a manifestation of the chemistry. I would consider it far more likely that the relatively modest pattern that does exist (noting that the vertical axes in figure 5d are not scaled from zero) is caused by the transport of regional pollution or marine air into the area during the day and concentrations decreasing at night through dry deposition in the nocturnal inversion layer, consistent with the wind speeds. It may also be that an analogous process, driven by dynamical rather than chemical processes, may be occurring for the highly oxidised SOA.

The third paragraph on page 8 makes a lot of very speculative statements concerning mechanisms that I do not consider to be adequately supported by the evidence and in certain cases implicitly rely on assumptions I do not consider safe. Not least of which is the tendency, as described in the general comments, to implicitly treat the factors represent discrete chemical types. But beyond this, I find the following lines of discussion to be problematic:

1) By using sulphate as a point of comparison for OOA, there is an intrinsic assumption that the precursors have similar sources on a regional level. While this is often the case at many sites, there are locations around the world (e.g. London) that do not conform to this behaviour. More evidence should be presented here that this is the case before drawing inferences concerning the chemical behaviour of these.

2) OOA-1 is described as being bimodal, however to my eyes, the trace in figure 5g simply has one very broad mode in its diurnal profile. Regardless, I fail to see how this constitutes evidence that there are different mechanisms in play in the morning versus the afternoon, so additional evidence to support this must be presented. Also see the comment concerning the treatment of SOA factors as discrete.

3) While a factor may correlate with nitrate, I consider it too much of a jump to assume that they are mechanistically related. I consider it more likely that they are formed over similar timescales and/or have similar thermodynamic properties.

Figure 5g: What are the red dashed lines?

Page 10: I would propose a much simpler explanation for the trends in the more highly oxidised material. If this is being formed regionally on longer timescales (as with sulphate), then this would already exist within the accumulation mode (which does not necessarily have to contain black carbon) and simply be advected in during the day. The less oxidised SOA formed more locally on shorter timescales could preferentially condense onto the BC particles through simple virtue of these Aiken mode particles presenting a larger overall surface area to the condensing gases. While it is hypothetically possible that there may be a mechanistic preference for material to condense onto certain surfaces because of their chemical nature, this would be a contentious statement to make and I do not see the data here as supportive of this notion.

Technical corrections:

Page 3, line 15: Which 42i was fitted with the photolytic converter? I assume the NOx instrument, but this should be explicit

Page 5, line 5: If the particle beam was found to be wider than the laser beam, this means that some particles may undergo incomplete vaporisation at the edges of the laser, which would in turn bias the measurement towards the coatings rather than the cores (see http://cires.colorado.edu/jimenez-group/UsrMtgs/UsersMtg16/JDASPAMSfocusing.pdf). This should be added as a caveat.

---

## Referee Comment (RC2) · Anonymous Referee #2 · 1 Sep 2017

Lee et al. investigated the mixing state of BC and the chemical characteristics of its coatings near vehicle emissions. They found that substantial formation of SOA coatings on BC particles was due to photochemistry in the afternoon, whereas POA was strongly associated with BC from fresh vehicular emissions in the morning rush hours. The paper potentially has significant implications to the formation of SOA coatings on BC in many urban environments, so is well within the scope for ACP. The paper is very interesting although some clarifications regarding the data analysis are required. The paper can be recommended for publication after the following questions are addressed.

General comments:

In Line 8-11, Page 5, the authors mentioned that CE for rBC particles increased (or rBC particle beam width decreases) with coating thickness. Although the variation of CE will not impact calculations of $R_{BC}$, it will certainly influence the calculation of NR-$PM_{rBC}$/NR-PM, which refers to the proportion of internally mixed aerosols. The authors should evaluate the impact of variable CEs on the calculation of NR-$PM_{rBC}$/NR-PM. In addition, gas-phase $CO_2$ will contribute to the $CO_2^+$ signal and thus influence the chemical composition of OA (Aiken et al., 2007;Aiken et al., 2008). Was the contribution of gas-phase $CO_2$ to $CO_2^+$ signal corrected in this study? This information was missing in the manuscript.

Specific comments:

line 19-20, page 5: The data for SP-AMS on July 21are not available, please give some clarifications.

line 26, page 5: What are the uncertainties of these items in Figure 2?

line 26-31, page 5: During the daytime from July 21 to 22, mass loadings of rBC and nitrate peaked in the afternoon on July 21 and in the morning on July 22 while the $NO_x$ mixing ratios kept at relatively low levels. Does this indicate that other sources of rBC existed? Should the data on these two days be excluded from the calculation? Also, the poor correlation between $NO_x$ and nitrate on these days may indicate formation mechanisms of particulate $NH_4NO_3$ other than the OH radicals oxidation of $NO_2$.

line 37-39, page 5: Although sulfate followed the diurnal patterns of nitrate in the morning, it does not really mean its significant formation from photochemistry. Is it possible that regional transportation contributed to this pattern? Is there any other evidence such as the pattern of the precursor gas $SO_2$ to draw this statement? Figure 1 shows that there are quite some changes in wind directions/speed during the hot periods.

Page 6. The authors have indicated that $C_x^+$ signals can be derived from standard rBC. However, in the PMF analysis of the NR-PMrBC, $C_x^+$ were used to identify an rBC rich

factor. I am somewhat confused here. Are data on Figure 3 and discussions on Page 6 (first paragraph) are for the whole particle or the coating? If they are just for the coating, why do these $C_x^+$ exist and how would we know if they are not from rBC?

Line 20-30, page 6. What is the major source of OOA-2? It does not seem to be the result of further reactions of OOA-1. The authors also discard the possibility of regional transport although it was mentioned in the conclusion later.

Line 33, Line 6. It is a bit awkward to me that $R_{BC}$ is called "thickness", given that it is a mass ratio.

line 6-7, page 7: The term POA may confuse the readers that COA is also included.

line 4-11, page 8: As shown in Figure S6, COA exhibited two peaks, one was around 17:00-18:00 similar to that of VOOA and the other larger peak was at midnight. This trend is largely different from typical diurnal patterns of COA that show a strong peak during dinner time. Also, large fractions of oxidized fragments were observed for $m/z$ 28, 42 and 55 in COA mass spectra. Could COA be already oxidized and not primary in origin?

Line 15, page 8. Can organic sulfur compounds be formed in relation to VOOA? Huang et al. (2015) estimated OS lower bounds based on AMS measurements in HK.

At times, I am confused what exactly rBC means. Is it the refractory BC without OA or including the OA (coating) on the BC? For example, line 8 on page 7, it says "OOA-1 contributed up to ~60 wt% of rBC mass…" This may be related to my earlier question on the nature of Cx+. On the other hand, terminology like rBC rich OA factor seems to imply a different definition of rBC. Figure 4 also uses "total rBC".

A conclusion is that the OOA in coatings of rBC are less oxygenated than those in VOOA. Just a speculation, would it be possible that the OOA in rBC condensed onto HOA coated rBC which promote partitioning of less oxygenated species while VOOA condensed on more hydrophilic particles?

Technical comments:

line 18, page 1: "missions" should be "emissions".

Reference:

Aiken, A. C., DeCarlo, P. F., and Jimenez, J. L.: Elemental Analysis of Organic Species with Electron Ionization High-Resolution Mass Spectrometry, Analytical Chemistry, 79, 8350-8358, 10.1021/ac071150w, 2007.

Aiken, A. C., DeCarlo, P. F., Kroll, J. H., Worsnop, D. R., Huffman, J. A., Docherty, K. S., Ulbrich, I. M., Mohr, C., Kimmel, J. R., Sueper, D., Sun, Y., Zhang, Q., Trimborn, A., Northway, M., Ziemann, P. J., Canagaratna, M. R., Onasch, T. B., Alfarra, M. R., Prevot, A. S. H., Dommen, J., Duplissy, J., Metzger, A., Baltensperger, U., and Jimenez, J. L.: O/C and OM/OC Ratios of Primary, Secondary, and Ambient Organic Aerosols with High-Resolution Time-of-Flight Aerosol Mass Spectrometry, Environ Sci Technol, 42, 4478-4485, 10.1021/es703009q, 2008.

---

## Author Comment (AC1) · 4 Nov 2017

**Reviewer 1**

**General comments**

This paper compares SP-AMS and AMS data at an urban location in the US to infer details concerning the behaviour of black carbon aerosols and SOA formation in this environment. This is partly facilitated by the dataset exhibiting a strong diurnal profile, meaning that traits can be reported with decent statistical power, in spite of the sampling period being relatively short. While the quantitative accuracy of the SP-AMS may be questionable (many of the RIEs and CEs are simply assumed rather than calibrated here), this does not form the major part of the science. This paper is well written and has the potential to provide some useful insight relevant to ACP, however I feel that the discussion side of the paper needs more work. Specifically, the authors appear to be making two unsafe assumptions often inappropriately made when analysing data of this nature; firstly, they seem to be assuming that the patterns are governed by sources and in situ processes (neglecting advection), akin to a box model and second, they seem to be treating the PMF factors as discrete chemical entities. While I'm sure the authors are aware of the issues here, this is not currently reflected in the text and a few conclusions are made that I would regard as unsafe. I recommend that this be published subject to major corrections.

Response: We thank for the reviewer's comments. We agree with the reviewer that the original version did not clearly reflect the potential impacts of advection on our diurnal observations. Furthermore, as the reviewer suggested in the specific comments, OOA factors identified by PMF analysis are combined in the revised version in order to provide more objective interpretation and comparison between the two AMS measurements. The manuscript has been revised accordingly throughout the discussion as highlighted in the specific comments below.

**Specific comments**

There an issue that I can see regarding the conclusions of the comparison of organic factors from the two AMSs in that the authors appear to be treating the PMF-derived SOA factors as discrete chemical types. While this could hypothetically be the case, it is far more likely that when applying the simple PMF data model to a chemically complex system such as this, the two SOA factors for each instrument represent the mathematical endpoints of a distribution of data, within the non-negativity constraint imposed. This being the case, a direct comparison of the two instruments' factors may be problematic; given that it is known their mass spectral responses to different organic materials are fundamentally different, their respective factorisations could end up being very different in character for the same aerosol simply through virtue of how the spectra are distributed in multivariate space and what endpoints are converged on by the solver. As a suggestion, a more objective comparison of the two datasets might be achieved if the primary factors (determined using PMF) are subtracted from the data matrices then metrics such as elemental ratios derived from what is left (assuming it is all secondary), e.g. as new points on figure 7. See https://www.atmos-chem-phys.net/15/6351/2015/.

Response:

We agree with the reviewer that comparison of total OOA measured by the two instruments may be more objective. Similar to Figure 6 in the original version, we have included the comparison of total OOA measured by both AMS instruments (i.e. $SOA_{NR-PM}$ = NOOA + VOOA for HR-ToF-

AMS and SOA$_{rBC}$ = OOA-1 and OOA-2 for SP-AMS) in Figure 7 as suggested by the reviewer. Note that signals of organic fragments at m/z 30 and 46 were removed from the updated PMF input due to the significant interferences of nitrate fragments to these organic fragments (Chen et al., submitted). With the updated results of PMF analysis (see Figure S6 in SI), Figures 6 and 7 have been modified accordingly in the revised manuscript. The results presented in Figure 6 are similar to those reported in the original version. Discussion for Figure 7 has been modified as shown below:

[revised manuscript text omitted]

An entire line of discussion – comparing the behaviour of organics to sulphate – I consider to be deeply flawed in the way it has been approached here. Outside of aqueous-phase processing (which is almost certainly not happening here, given the low humidities), the formation lifetime of sulphate from the gas phase oxidation of sulphur dioxide tends to occur on the timescale of weeks (far beyond the timescale where NOy/NOx is a useful indicator of photochemistry), which means that a diurnal pattern observed here is extremely unlikely to be a manifestation of the chemistry. I would consider it far more likely that the relatively modest pattern that does exist (noting that the vertical axes in figure 5d are not scaled from zero) is caused by the transport of regional pollution or marine air into the area during the day and concentrations decreasing at night through dry deposition in the nocturnal inversion layer, consistent with the wind speeds. It may also be that an analogous process, driven by dynamical rather than chemical processes, may be occurring for the highly oxidised SOA.

Response:

The sentence has been revised as shown below to address the possibility of regional transport of sulfate during the day.

Page 6: "Sulfate remained in low concentrations with relatively modest pattern (see also Figure 5d) potentially caused by the transport of regional pollution into the area during the day."

Furthermore, the above information has been integrated to the discussion regarding the potential sources of VOOA (i.e. more oxidized OOA materials) as shown below:

Page 8: "$SOA_{NR-PM}$ mass concentrations started dropping substantially at ~18:00-19:00, which was about 1-2 hours delay compared to $SOA_{rBC}$ (Figure 5g). This observable delay corresponded to the time of increasing sulfate levels in NR-PM, implying potential sources and formation pathways of VOOA (i.e., the major component of $SOA_{NR-PM}$ within that period) that might be related to the regional transport of aged particles. This possible explanation is consistent with the observation that VOOA represented more oxidized OOA materials (i.e. more aged) and that the strongest

average wind speed was observed at around 18:00-19:00 (Figure 2c). Single particle measurements using the light scattering module of the HR-ToF-AMS also suggests internal mixing of sulfate and highly oxidized OOA materials in NR-PM (Chen et al., 2017)."

The third paragraph on page 8 makes a lot of very speculative statements concerning mechanisms that I do not consider to be adequately supported by the evidence and in certain cases implicitly rely on assumptions I do not consider safe. Not least of which is the tendency, as described in the general comments, to implicitly treat the factors represent discrete chemical types. But beyond this, I find the following lines of discussion to be problematic:

1) By using sulphate as a point of comparison for OOA, there is an intrinsic assumption that the precursors have similar sources on a regional level. While this is often the case at many sites, there are locations around the world (e.g. London) that do not conform to this behaviour. More evidence should be presented here that this is the case before drawing inferences concerning the chemical behaviour of these.

Response:

In this study, the HR-ToF-AMS was equipped with the light scattering module (LS-AMS), and thus single particle data was available to investigate the aerosol mixing state (Chen et al., submitted). By performing cluster analysis on the single particle data, it can be found that a portion of more- oxidized SOA materials were internally mixed with sulfate-rich particles. This provides direct evidence that at least a portion of VOOA materials were associated with the regional transport of sulfate. The aspect of internal mixing between VOOA and sulfate is addressed in the revised paragraph in our earlier response.

2) OOA-1 is described as being bimodal, however to my eyes, the trace in figure 5g simply has one very broad mode in its diurnal profile. Regardless, I fail to see how this constitutes evidence that there are different mechanisms in play in the morning versus the afternoon, so additional evidence to support this must be presented. Also see the comment concerning the treatment of SOA factors as discrete.

Response:

The sentences regarding bimodal pattern of OOA-1 have been removed in the revised version. Instead, we focus on discussing the diurnal patterns of total OOA measured by the two AMS measurements (see the response for the first specific comments), and the following sentence has been added for general description of OOA-1 factor in Section 3.2.

Page 6: "OOA-1 represented less oxygenated SOA components (O/C = 0.53) with 13.4 wt% of rBC content, and its concentration started increasing in the morning coincident with nitrate and ammonium (i.e., peaks at ~10:00-11:00, Figure 2a and 3g) and sustained at relatively constant levels until ~15:00–16:00."

3) While a factor may correlate with nitrate, I consider it too much of a jump to assume that they are mechanistically related. I consider it more likely that they are formed over similar timescales and/or have similar thermodynamic properties.

Response:

After modifying the discussion in this paragraph, the related sentences have been removed in the revised version.

Figure 5g: What are the red dashed lines?

Response:

The red dashed lines have been removed in Figure 5 due to the modified discussion for comparing SP-AMS and HR-ToF-AMS measurements.

Page 10: I would propose a much simpler explanation for the trends in the more highly oxidised material. If this is being formed regionally on longer timescales (as with sulphate), then this would already exist within the accumulation mode (which does not necessarily have to contain black carbon) and simply be advected in during the day. The less oxidised SOA formed more locally on shorter timescales could preferentially condense onto the BC particles through simple virtue of these Aiken mode particles presenting a larger overall surface area to the condensing gases. While it is hypothetically possible that there may be a mechanistic preference for material to condense onto certain surfaces because of their chemical nature, this would be a contentious statement to make and I do not see the data here as supportive of this notion.

Response:

The conclusion has been modified based on the suggestion from the reviewer. Even though we cannot provide direct evidence of a mechanistic preference for material to condense onto certain surfaces, we have changed our tone in the revised version to highlight this possibility for future research.

Page 10: "One of the possibilities is that $SOA_{rBC}$ formed more locally on shorter timescales (e.g., photo-oxidation of anthropogenic volatile organic compounds (VOCs) near traffic emissions) could preferentially condense onto rBC particles in Aiken mode that can provide a larger overall surface area to the condensing gases compared to other existing background particles. The more oxidized OOA materials formed regionally on longer timescales (e.g., a fraction of VOOA that were largely externally mixed with rBC) under conditions with relatively low concentrations of (or without) rBC particle could be advected to the sampling region during the day. Furthermore, atmospheric dilution of traffic emissions can modify the chemical compositions and concentrations of SOA precursors and seed particles so that the formation of secondary coatings on rBC might be less efficient and chemically different under diluted conditions (e.g., after boundary layer break up and mixing with air masses from residual layer). This may partially explain the formation of

VOOA in SOA$_{NR-PM}$ through local photochemistry. There may also be a mechanistic preference for material to condense onto certain surfaces because of their chemical nature (e.g., hydrophobic coating (e.g. HOA and soot surface functionality) but future research efforts are required to explore this possibility further. "

**Technical corrections:**

Page 3, line 15: Which 42i was fitted with the photolytic converter? I assume the NOx instrument, but this should be explicit

Response: Yes, the photolytic converter was used for NO$_x$ measurement. The sentence has been revised as following.

Page 3: "The heated molybdenum converter in the 42i nitrogen oxide analyzer was replaced by an UV-LED photolytic NO$_2$ converter (Air Quality Design) for NO$_x$ measurement."

Page 5, line 5: If the particle beam was found to be wider than the laser beam, this means that some particles may undergo incomplete vaporisation at the edges of the laser, which would in turn bias the measurement towards the coatings rather than the cores (see http://cires.colorado.edu/jimenezgroup/UsrMtgs/UsersMtg16/JDASPAMSfocusing.pdf). This should be added as a caveat.

Response: We agree that such measurement uncertainty should be included in the experimental section. The following sentence has been added to the revised manuscript (Page 5, lines 14-20):

Page 5: "The applied CE may be therefore less relevant for the time with high ambient SOA loading (i.e., high NR-PM$_{rBC}$/rBC mass ratio, or R$_{BC}$), leading to over quantification of the SOA components in NR-PM$_{rBC}$ by at most 40% due to this uncertainty. Furthermore, a wider particle beam than the laser beam implies that some rBC-containing particles may undergo incomplete vaporization at the edges of the laser vaporizer, which would in turn bias the measurement towards the coatings rather than the rBC cores (see unpublished data from, http://cires.colorado.edu/jimenezgroup/UsrMtgs/UsersMtg16/JDASPAMSfocusing.pdf). Since the CE of 0.6 was primarily determined for rBC, this phenomenon may further increase the degree of over quantification of NR-PM$_{rBC}$. Overall, the values of NR-PM$_{rBC}$ reported in this work likely represent their upper limits."

**Reviewer 2**

Lee et al. investigated the mixing state of BC and the chemical characteristics of its coatings near vehicle emissions. They found that substantial formation of SOA coatings on BC particles was due to photochemistry in the afternoon, whereas POA was strongly associated with BC from fresh vehicular emissions in the morning rush hours. The paper potentially has significant implications to the formation of SOA coatings on BC in many urban environments, so is well within the scope for ACP. The paper is very interesting although some clarifications regarding the data analysis are required. The paper can be recommended for publication after the following questions are addressed.

Response: We thank for the reviewer of the positive comments.

**General comments:**

In Line 8-11, Page 5, the authors mentioned that CE for rBC particles increased (or rBC particle beam width decreases) with coating thickness. Although the variation of CE will not impact calculations of RBC, it will certainly influence the calculation of NR-PMrBC/NR-PM, which refers to the proportion of internally mixed aerosols. The authors should evaluate the impact of variable CEs on the calculation of NR-PMrBC/NR-PM.

Response:

We agree with the reviewer that we need to provide some information on the measurement uncertainties of NR-PM$_{rBC}$ in this work. As described in the manuscript and the literature, there are three major uncertainties for NR-PM$_{rBC}$ quantification: 1) CE of rBC-containing particles due to incomplete overlapping of particle and laser beams (Willis et al., 2014, page 5 lines 14-17 in the revised version), 2) IE calibration approach for laser vaporization scheme used in SP-AMS (Lee et al., 2015, page 5, lines 3-5), and 3) partial vaporization of particles at the edge of laser vaporizer that bias the measurement towards the coating rather than the rBC cores (unpublished data, http://cires.colorado.edu/jimenezgroup/UsrMtgs/UsersMtg16/JDASPAMSfocusing.pdf).

With the current estimation of CE=0.6 for all rBC-containing particles (point 1), NR-PM$_{rBC}$ of aged particles (i.e., those with high R$_{BC}$) may be over quantified by at most 40% (Willis et al., 2014). Lee et al. (2015) has illustrated that applying the standard mass-based IE calibration approach (point 2) may lead to over quantification of NR-PM$_{rBC}$ (i.e. apparent increased sensitivity to NR-PM$_{rBC}$ vaporized in laser vaporizer) for different types of particles measured by a SP-AMS equipped with dual vaporizers, and such uncertainty is likely instrument and vaporizer configuration dependent. Lastly, partial vaporization at the edge of laser vaporizer (point 3) results in a higher CE for NR-PM$_{rBC}$ than the rBC cores, leading to further over quantification of NR-PM$_{rBC}$ when CE correction factor of 0.6 is applied for all aerosol components. However, quantification of the overall uncertainties due to these major factors is not straightforward, and their importance have been clearly highlighted in the 17[th] AMS users meeting in 2016 after this field study (see summary slides prepared by Tim Onash from http://cires1.colorado.edu/jimenezgroup/UsrMtgs/UsersMtg17/2015%20SP-AMS%20upate%20presentation%20-%20Onasch.pdf).

Even though the uncertainty of NR-PM$_{rBC}$ measurement cannot be accurately quantified in this work, the three major uncertainties lead to over quantification of NR-PM$_{rBC}$ in general, especially for those particles with high R$_{BC}$ (i.e. aged particles dominated by secondary inorganic and organic components). That means the NR-PM$_{rBC}$/NR-PM ratios reported in this work likely represent the upper limit. Including this additional information in the revised manuscript would not affect one of our major conclusions that only a small fraction of secondary organic and inorganic species were estimated to be internally mixed with rBC on average, implying that rBC is unlikely the major condensation sinks of SOA in this study.

Based on the comments from both reviewers, the original argument has been modified in the revised manuscript as shown below:

Page 5: "The applied CE may be therefore less relevant for the time with high ambient SOA loading (i.e., high NR-PM$_{rBC}$/rBC mass ratio, or R$_{BC}$), leading to over quantification of the SOA components in NR-PM$_{rBC}$ by at most 40% due to this uncertainty. Furthermore, a wider particle beam than the laser beam implies that some rBC-containing particles may undergo incomplete vaporization at the edges of the laser vaporizer, which would in turn bias the measurement towards the coatings rather than the rBC cores (see unpublished data from, http://cires.colorado.edu/jimenezgroup/UsrMtgs/UsersMtg16/JDASPAMSfocusing.pdf). Since the CE of 0.6 was primarily determined for rBC, this phenomenon may further increase the degree of over quantification of NR-PM$_{rBC}$. Overall, the values of NR-PM$_{rBC}$ reported in this work likely represent their upper limits."

In addition, gas-phase CO2 will contribute to the CO2+ signal and thus influence the chemical composition of OA (Aiken et al., 2007; Aiken et al., 2008). Was the contribution of gas-phase CO2 to CO2+ signal corrected in this study? This information was missing in the manuscript.

Response:

The contribution of gas-phase $CO_2$ to organic fragment $CO_2^+$ was determined by measuring the gas-phase $CO_2$ to $N_2$ ratio of particle-free ambient air. Such information has been added to the revised manuscript as shown below:

Page 4: "In particular, the average contribution of gas-phase $CO_2$ to $CO_2^+$ organic fragment in particle phase was determined (i.e., $CO_2$-to-$N_2$ ratio) based on the measurements of particle-free ambient air (i.e., at least 10 min per day) throughout the sampling period."

**Specific comments:**

Line 19-20, page 5: The data for SP-AMS on July 21 are not available, please give some clarifications.

Response: There were a few SP-AMS technical issues to be fixed from around July 20 18:00 to July 21 12:00. The following sentence has been added to the revised manuscript.

Page 5: "The SP-AMS was under maintenance from 20 July 18:00 to 21 July 12:00."

Line 26, page 5: What are the uncertainties of these items in Figure 2?

Response:

The major purpose of Figure 2 is to illustrate the diurnal trend of each species during the hot period. We have added a new figure (Figure S7) in the supplementary information to show the variability (i.e. average ± one standard deviation) of these items. The caption of Figure 2 has been modified to direct the reader to Figure S7 if they are interested in the variability of these measurements.

[Figure]

Figure S7: Diurnal cycles of (a) rBC, NR-PM$_{rBC}$ components, including (b) Organics, (c) nitrate, (d) ammonium, (e) sulfate and (f) chloride, (g) ozone, (h) -log(NO$_x$/NO$_y$), (i) R$_{BC}$, (j) Org/rBC ratio, (k) O/C and (l) H/C of organic coating. The data points represent average values and the error bars represent one standard deviation.

Line 26-31, page 5: During the daytime from July 21to 22, mass loadings of rBC and nitrate peaked in the afternoon on July 21 and in the morning on July 22 while the NOx mixing ratios kept at relatively low levels. Does this indicate that other sources of rBC existed? Should the data on these two days be excluded from the calculation? Also, the poor correlation between NOx and nitrate on these days may indicate formation mechanisms of particulate NH4NO3 other than the OH radicals oxidation of NO2.

Response:

Thanks for addressing this interesting point. rBC can be emitted from any combustion sources but we did not identify combustion emissions (such as biomass burning) other than traffic from the PMF analysis of both HR-ToF-AMS and SP-AMS measurements. Furthermore, SP-AMS data is not available on the morning of July 21 so that we cannot comment whether the nitrate concentrations would have peaked in the morning on this day. We have tried to exclude that short period of time to calculate the diurnal cycles of individual species. The changes in diurnal patterns and concentrations are only minimal and thus we decide to keep the original period for all the subsequent calculation in this work. We agree that the elevated NH$_4$NO$_3$ concentrations within that short period of time might be due to regional transport and/or produced through mechanisms other than OH radical oxidation of NO$_2$. Since this paper aims to present the general observations/patterns of aerosol species within the entire hot period, we decided to not discuss this individual event in the revised manuscript in order to avoid any potential confusion to readers.

Line 37-39, page 5: Although sulfate followed the diurnal patterns of nitrate in the morning, it does not really mean its significant formation from photochemistry. Is it possible that regional transportation contributed to this pattern? Is there any other evidence such as the pattern of the precursor gas SO2 to draw this statement? Figure 1 shows that there are quite some changes in wind directions/speed during the hot periods.

Response:

Reviewer 1 also points out this possibility. The sentence has been revised as shown below to address the possibility of regional transport of sulfate during the day.

Page 6: "Sulfate remained in low concentrations with a relatively modest pattern (see also Figure 5d) potentially caused by the transport of regional pollution into the area during the day"

Page 6. The authors have indicated that Cx+ signals can be derived from standard rBC. However, in the PMF analysis of the NR-PMrBC, Cx+ were used to identify an rBC rich factor. I am

somewhat confused here. Are data on Figure 3 and discussions on Page 6 (first paragraph) are for the whole particle or the coating? If they are just for the coating, why do these Cx+ exist and how would we know if they are not from rBC?

Response:

$C_x^+$ ions (from rBC) were included in the PMF analysis to identify potential mass contributions of rBC to each PMF factor (i.e., each PMF factor consists of both rBC and OA components). This approach has been used in our previous studies (Willis et al., 2016, and Lee et al., 2015, 2016). Signals of $C_x^+$ fragments can be used to calculate rBC mass (or mass fraction) in each PMF factor whereas other organic fragments represent different types of organic coating. To avoid potential confusion here, we modify a sentence in the experimental section as following:

Page 4: "Positive matrix factorization (PMF) was performed to investigate the potential sources and characteristics of rBC and organic aerosol components. Signals of $C_x^+$ fragments from rBC were included in the PMF analysis so that mass fraction of rBC and organic components can be calculated for each PMF factor (Lee et al., 2015; 2016;Willis et al., 2016)."

Line 20-30, page 6. What is the major source of OOA-2? It does not seem to be the result of further reactions of OOA-1. The authors also discard the possibility of regional transport although it was mentioned in the conclusion later.

Response:

While OOA-2 was likely due to local photochemistry, the possibility of regional transport of OOA-2 has been pointed out in the discussion as shown below:

Page 6: "OOA-2 represented a more oxygenated fraction of SOA coating (O/C = 0.62) with the maximum mass loadings observed at ~15:00-16:00 and its diurnal pattern matched well with Org/rBC ratios (Figures 2b and 3h). The diurnal pattern of OOA-2 indicates the importance of local photochemistry for OOA-2 production in the afternoon but the contribution of regional transport to OOA-2 cannot be completely ruled out. Given that rBC accounted for only 0.5 wt% of OOA-2 (i.e. much lower than other PMF factors), OOA-2 could represent SOA materials generated through local photochemistry and/or formed regionally under conditions with relatively low rBC particle concentrations."

Line 33, Line 6. It is a bit awkward to me that RBC is called "thickness", given that it is a mass ratio.

Response:

The term "coating thickness" has been removed from the heading of Section 3.2.

Line 6-7, page 7: The term POA may confuse the readers that COA is also included.

Response:

The term "POA" has been changed to "traffic-related POA" to avoid confusion.

Line 4-11, page 8: As shown in Figure S6, COA exhibited two peaks, one was around 17:00-18:00 similar to that of VOOA and the other larger peak was at midnight. This trend is largely different from typical diurnal patterns of COA that show a strong peak during dinner time. Also, large fractions of oxidized fragments were observed for $m/z$28, 42 and 55 in COA mass spectra. Could COA be already oxidized and not primary in origin?

Response:

The updated results of PMF analysis is shown in Figure S6 in SI. Signals of organic fragments at m/z 30 and 46 were removed from the PMF input due to the significant interferences of nitrate fragments to these organic fragments (Chen et al., submitted). Figure S6b shows the updated mass spectrum of COA, which has O/C and H/C ratios of 0.12 and 1.83 respectively.  Even though COA peaked at around 18:00-19:00, the updated values of O/C and H/C ratios suggests the primary nature of the observed COA. The mass spectral characteristics of COA (i.e. high m/z 55/57 and m/z 41/43 ratios and relatively low contributions of oxygenated fragments to total organic) are also consistent to relatively fresh COA observed from previous studies (e.g., Mohr et al., 2012, Kaltsonoudis et al., 2017). The detail of PMF factors determined from NR-PM has been discussed in our companion study (Chen et al., submitted), and thus is not the focus of this work. However, it is worth noting that high levels of COA factor over mid-night is observed previously at a rural site (Dall'Osto et al., 2015), suggesting that the nature and origins of COA are complex and can include more than food cooking.

Line 15, page 8. Can organic sulfur compounds be formed in relation to VOOA? Huang et al. (2015) estimated OS lower bounds based on AMS measurements in HK.

Response:

Formation of organic sulfur compounds is certainly an interesting topic in aerosol chemistry. Concentrations of organic sulfur compounds in NR-PM was not correlated well with VOOA in this field study, and this information has been reported in our companion study (Chen et al., submitted). The results of comprehensive HR-ToF-AMS data analysis can also be found in that paper as well.

At times, I am confused what exactly rBC means.  Is it the refractory BC without OA or including the OA (coating) on the BC?  For example, line 8 on page 7, it says "OOA-1 contributed up to ~60 wt% of rBC mass…"  This may be related to my earlier question on the nature of Cx+. On the

other hand, terminology like rBC rich OA factor seems to imply a different definition of rBC. Figure 4 also uses "total rBC".

Response:

rBC is refractory black carbon without OA. This technical term has been used to describe black carbon measured by SP-AMS and SP2. As mentioned in our previous response, $C_x^+$ ions (from rBC) were included in PMF analysis to identify potential mass contributions of rBC to each PMF factor (i.e., each PMF factor consists of both rBC and OA components). This approach has been used in our previous field studies (Willis et al., 2016, and Lee et al., 2015, 2016). Signals of $C_x^+$ fragments can be used to calculate rBC mass (or mass fraction) for each PMF factor whereas other organic fragments represent different types of organic coating.

For example, rBC accounts for ~14 wt% and ~44 wt% of HOA-rich and rBC-rich factors, respectively (Page 6). In other words, OA accounts for ~86 wt% and 56 wt% of HOA-rich and rBC-rich factors, respectively. Similarly, based on the mass fraction of rBC determined in each PMF factor, the contribution of each PMF factor to total rBC concentrations can be calculated as shown in Figure 4d.

A conclusion is that the OOA in coatings of rBC are less oxygenated than those in VOOA. Just a speculation, would it be possible that the OOA in rBC condensed onto HOA coated rBC which promote partitioning of less oxygenated species while VOOA condensed on more hydrophilic particles?

Response:

This speculation is possible. However, as pointed out by Reviewer 1, we cannot provide direct evidence of a mechanistic preference for material to condense onto certain surfaces, we have changed our tone in the revised version to highlight this possibility for future research.

Page 10: "There may be also a mechanistic preference for material to condense onto certain surfaces because of their chemical nature (e.g., hydrophobic coating (e.g. HOA and soot surface functionality) but more research efforts are required to explore this possibility."

**Technical comments:**

Line 18, page 1: "missions" should be "emissions".

Response: The typo has been corrected.

---

## Author Comment (AC2) · 4 Nov 2017

The comment was uploaded in the form of a supplement:
https://www.atmos-chem-phys-discuss.net/acp-2017-665/acp-2017-665-AC2-supplement.pdf

---

## Author Response (AR1)

**Reviewer 1**

**General comments**

This paper compares SP-AMS and AMS data at an urban location in the US to infer details concerning the behaviour of black carbon aerosols and SOA formation in this environment. This is partly facilitated by the dataset exhibiting a strong diurnal profile, meaning that traits can be reported with decent statistical power, in spite of the sampling period being relatively short. While the quantitative accuracy of the SP-AMS may be questionable (many of the RIEs and CEs are simply assumed rather than calibrated here), this does not form the major part of the science. This paper is well written and has the potential to provide some useful insight relevant to ACP, however I feel that the discussion side of the paper needs more work. Specifically, the authors appear to be making two unsafe assumptions often inappropriately made when analysing data of this nature; firstly, they seem to be assuming that the patterns are governed by sources and in situ processes (neglecting advection), akin to a box model and second, they seem to be treating the PMF factors as discrete chemical entities. While I'm sure the authors are aware of the issues here, this is not currently reflected in the text and a few conclusions are made that I would regard as unsafe. I recommend that this be published subject to major corrections.

Response: We thank for the reviewer's comments. We agree with the reviewer that the original version did not clearly reflect the potential impacts of advection on our diurnal observations. Furthermore, as the reviewer suggested in the specific comments, OOA factors identified by PMF analysis are combined in the revised version in order to provide more objective interpretation and comparison between the two AMS measurements. The manuscript has been revised accordingly throughout the discussion as highlighted in the specific comments below.

**Specific comments**

There an issue that I can see regarding the conclusions of the comparison of organic factors from the two AMSs in that the authors appear to be treating the PMF-derived SOA factors as discrete chemical types. While this could hypothetically be the case, it is far more likely that when applying the simple PMF data model to a chemically complex system such as this, the two SOA factors for each instrument represent the mathematical endpoints of a distribution of data, within the non-negativity constraint imposed. This being the case, a direct comparison of the two instruments' factors may be problematic; given that it is known their mass spectral responses to different organic materials are fundamentally different, their respective factorisations could end up being very different in character for the same aerosol simply through virtue of how the spectra are distributed in multivariate space and what endpoints are converged on by the solver. As a suggestion, a more objective comparison of the two datasets might be achieved if the primary factors (determined using PMF) are subtracted from the data matrices then metrics such as elemental ratios derived from what is left (assuming it is all secondary), e.g. as new points on figure 7. See https://www.atmos-chem-phys.net/15/6351/2015/.

**Response:**

We agree with the reviewer that comparison of total OOA measured by the two instruments may be more objective. Similar to Figure 6 in the original version, we have included the comparison of total OOA measured by both AMS instruments (i.e.  $SOA_{NR-PM} = NOOA + VOOA$  for HR-ToF-

AMS and  $SOA_{rBC} = OOA-1$  and OOA-2 for SP-AMS) in Figure 7 as suggested by the reviewer. Note that signals of organic fragments at m/z 30 and 46 were removed from the updated PMF input due to the significant interferences of nitrate fragments to these organic fragments (Chen et al., submitted). With the updated results of PMF analysis (see Figure S6 in SI), Figures 6 and 7 have been modified accordingly in the revised manuscript. The results presented in Figure 6 are similar to those reported in the original version. Discussion for Figure 7 has been modified as shown below:

Page 9: "The elemental ratios (O/C and H/C) extracted from the SP-AMS and HR-ToF-AMS mass spectra of oxidized organic species have been shown to be different. Canagaratna et al. (2015b) reported that the SP-AMS O/C and H/C values differ from their corresponding HR-ToF-AMS values by factors of 0.83 and 1.16, respectively, based on the laboratory analysis of chemical standards, including dicarboxylic acids, multifunctional acids and alcohols. These conversion factors are applied to the O/C and H/C ratios of NOOA, VOOA and SOANR-PM (i.e., mass-weighted values of NOOA and VOOA) (Chen et al., 2017) in order to perform a more meaningful comparison to our SP-AMS measurements in the Van Krevelen diagram (Figure 7). The elemental ratios of SOArBC and SOANR-PM are comparable to each other, well within the measurement uncertainties. Similar observations are obtained between OOA-2 and VOOA. In contrast, the O/C and H/C ratios of OOA-1 are rather different to the adjusted values of NOOA, VOOA and SOANR-PM, suggesting some chemical difference between OOA-1 and those SOA materials. This is also consistent with the fact that OOA-1 mass concentrations increased faster than other OOA materials in the morning. More field and laboratory data are required to validate and improve the empirical relationships proposed by Canagaratna et al., (2015b) and to understand the extent to which the observed differences are a result of true chemical differences versus explainable by differences in molecular fragmentation due to the different vaporization schemes used in the instruments."

Figure 7: Van Krevelen diagram: Red arrows indicate the changes in the elemental ratios of VOOA, NOOA factors and  $SOA_{NR-PM}$  measured by the HR-ToF-AMS after applying the correction factors for more oxygenated organic species proposed by Canagaratna et al. (2015b).

In addition, we have combined the diurnal variations of OOA factors as shown in the revised Figure 5g (i.e.,  $SOA_{rBC}$  vs.  $SOA_{NR-PM}$ ). Discussions for Figure 5g has been modified as shown below: